# COVID-19 Perceived Impact and Psychological Variables as Predictors of Unhealthy Food and Alcohol Consumption Trajectories: The Role of Gender and Living with Children as Moderators

**DOI:** 10.3390/ijerph18094542

**Published:** 2021-04-25

**Authors:** Camila Salazar-Fernández, Daniela Palet, Paola A. Haeger, Francisca Román Mella

**Affiliations:** 1Departamento de Psicología, Universidad de La Frontera, Temuco 4811230, Chile; camila.salazar@ufrontera.cl (C.S.-F.); daniela.palet@ufrontera.cl (D.P.); 2Departamento de Ciencias Biomédicas, Facultad de Medicina, Universidad Católica del Norte, Coquimbo 1781421, Chile; phaeger@ucn.cl

**Keywords:** pandemic, unhealthy behavior, longitudinal analysis, latent growth curve models, loneliness

## Abstract

The present study examines the trajectories of unhealthy food and alcohol consumption over time and considers whether perceived impact of COVID-19 and psychological variables are predictors of these trajectories. We ascertained whether these predictors are different in women vs. men and between women living with vs. without children. Data were collected through online surveys administered to 1038 participants from two universities (staff and students) in Chile, across five waves (July to October 2020). Participants provided information about their past-week unhealthy food and alcohol consumption and mental health. Using latent growth curve modeling analysis, we found that higher perceived health and interpersonal COVID-19 impact, younger age and lower depression symptoms were associated with more rapid increases over time in unhealthy food consumption. On the other hand, higher perceived COVID-19 economic impact and older age were associated with more rapid diachronic decreases in alcohol consumption. Gender and living with or without children, for women only, were moderators of these trajectories. This longitudinal study provides strong evidence identifying the multiple repercussions of COVID-19 and mental health factors on unhealthy food and alcohol consumption. These findings highlight the need for interventions aimed at minimizing the impact of the pandemic on unhealthy food and alcohol consumption over time.

## 1. Introduction

The prevalence of unhealthy food consumption and high alcohol consumption is growing at an alarming rate in the Americas [1]. Alcohol is the most used toxic substance worldwide and its harmful consumption causes several disabilities and even death, while obesity and overweight are the second leading cause of preventable deaths [2,3,4]. Together, they constitute significant risk factors for numerous non-communicable chronic diseases such as diabetes, cardiovascular disease and cancer [5]. These numbers are of particular concern in the light of several studies that have reported that unhealthy food and alcohol consumption has risen during the COVID-19 pandemic [6,7,8,9].

The SARS-CoV-2 virus has spread rapidly worldwide following its detection at the end of 2019 [10]. In March 2020, it was officially declared a pandemic by the WHO. Since then, active cases and deaths have increased worryingly around the world [11]. Due to the health emergency, several protective measures have been taken by the governments, including lockdowns, closure of workplaces, schools, public places and continuous advertising of stay at home and maintain social distancing messages.

Staying home and working from home during the pandemic have implications for daily life ranging from dealing with economic impairment (i.e., job/income loss) [12] and work-overload [13] to increased parenting demands (i.e., child care and homeschooling). In addition to all these abrupt changes in daily life, the pandemic has left many people without access to adaptive coping resources, such as social support from family and friends and access to recreational facilities, health and counseling services and places of worship [12,14]. Previous research has shown that changes in these lifestyle-behaviors during lockdowns, or when more severe social distancing restrictions are in place, are associated with adverse psychological effects [10,15] impacting mental health and wellbeing [16,17,18], alongside more stress [19] and anxiety [20]. More importantly, elevated stress has been associated with non-healthy coping mechanisms such as food overconsumption (i.e., increased motivation to eat, snacking, reduced diet quality) [21] and increased alcohol consumption [6,22].

Although the pandemic and lockdown directly impact all lives, prior research has identified women and women living with children as being among the most affected groups [18,23]. The roles of working women and working mothers before the pandemic were already stressful, overwhelming and lonely [24]. Recent evidence from Italy and the UK has shown that women, in reaction to the current changing circumstances, have experienced a dramatic increase in the amount of time they spend on their jobs, housework and childcare [25,26]. This struggle to balance work and family life during the pandemic is an important cause of psychological distress [27].

In sum, constraining daily routines and limited access to typical social or environmental reinforcement sources may increase the risk of engaging with non-healthy strategies to cope with the pandemic, such as eating more unhealthy food and drinking more alcohol than usual [6,7,8,9,28], especially for women. Although previous cross-sectional studies have found an increase in these high-risk behaviors [29,30,31], it is unclear what predicts these changes and, more importantly, if these behaviors are maintained over time. Since the pandemic has been found to increase levels of stress across the population, it will be necessary to evaluate changes over time in health-risk behaviors and their predictors, that are known to be affected by stress [32]. In this study, we aimed to assess three objectives: (1) whether unhealthy food and alcohol consumption trajectories have changed over time during the pandemic, (2) if variables such as age, perceived impact of COVID-19 and psychological variables (anxiety, depression, stress, loneliness) can predict these longitudinal changes and (3) whether these variables predicting the trajectories are moderated by gender and by living with or without children.

## 2. Materials and Methods

### 2.1. Participants and Procedure

University students and staff (academic and non-academic) of two Chilean universities were invited to participate in online surveys. Participants were informed of the nature of the study and signed an informed consent indicating their willingness to participate. The Scientific Ethics Committee from both universities approved the study. No material remuneration was provided in exchange for their participation.

The current study included 1038 participants who reported their alcohol and unhealthy food consumption behaviors over four waves during July (Wave 1, *n* = 1038), August (Wave 2, *n* = 509), September (Wave 3, *n* = 413) and October (wave 4, *n* = 475). See Table 1 for a description of participant characteristics.

### 2.2. Measures

Participants responded to online questionnaires designed to collect recent information regarding their health-related behaviors during the COVID-19 pandemic. Online questionnaires were sent using QuestionPro and completion took participants approximately 10 min. The present study only used the following questions/instruments to analyze specific unhealthy food and alcohol consumption behaviors during the pandemic.

### 2.3. Outcomes (Measured in Waves 1, 2, 3 and 4)

#### 2.3.1. Past-Week Alcohol Consumption

Participants were asked, “In the past week, how many drinks did you consume?”. Responses were scored on a scale from 0 to 20 drinks. Participants were provided with information about standard alcoholic drinks, using images and giving the volume of an equivalent amount of beer, wine and spirits. They had to report the number of drinks consumed yesterday and during the previous week (7 days). These seven measures (1 per day) were calculated as a total score. High scores indicate a greater weekly amount of alcohol consumed. Cronbach’s alpha of the scale was acceptable (*α* = 0.741).

#### 2.3.2. Past-Week Unhealthy Food Consumption

A 5-item question asked, “During the last week, on how many days have you consumed: fried meals (e.g., fried meat, fish, eggs, fries), sugary drinks (i.e., cola drinks, bottled juice), desserts or candies (e.g., ice cream, chocolate, candies, cakes, pastries), snacks (potato chips, chocolate bars, candy bar, cookies) and fast food (e.g., hamburgers, pizzas, hot dogs).” All these questions refer to foods that are highly processed and rich in saturated fat, sugar and sodium. Responses were scored from 0 to 7 days. Responses were summed to provide a total score indicating comfort food consumption. High scores indicated greater unhealthy food consumption. A reliability analysis showed acceptable levels (*α* = 0.768).

### 2.4. Predictors Measured in Wave 1

#### 2.4.1. Perceived COVID-19 Impact

For the following three items, participants were asked to think about how much the COVID-19 pandemic has affected them.

Perceived COVID-19 economic impact. A single item question asked: “COVID-19 has negatively affected my economic or employment situation.” Responses were scored on a 5-point scale: not at all (0), a little (1), some (2), quite a bit (3), a lot (4). Higher scores indicated a greater economic impact of COVID-19.

Perceived COVID-19 health impact. A single item question asked: “COVID-19 has negatively affected me or my loved ones’ health”. Responses were scored on a 5-point scale (not at all (0), a little (1), some (2), quite a bit (3), a lot (4). Higher scores indicated a greater health impact of the COVID-19.

Perceived COVID-19 interpersonal impact. A single item question asked: “COVID-19 has negatively affected my personal relations with family and friends”. Responses were scored on a 5-point scale (not at all (0), A little (1), some (2), quite a bit (3), a lot (4)). Higher scores indicated a greater interpersonal impact of COVID-19.

#### 2.4.2. Depression, Anxiety and Stress (DASS-21)

A 21-item (7 items each) abbreviated scale measuring each of these constructs was used in this study [33]. This scale, scored on a 4-point severity scale (from 0 to 3), estimates the presence and intensity of affective states experienced over the past week. Cronbach’s alpha for each scale ranged from good to excellent (Depression, *α* = 0.913, Anxiety, *α* = 0.891, Stress, *α* = 0.903). Items were tallied to provide a total score for depression, anxiety and stress.

#### 2.4.3. Loneliness

A 5-point item asked: “Since the beginning of the COVID-19, how frequently have you felt lonely?”. Responses were scored as: 0 = never; 1 = rarely; 2 = sometimes; 3 = frequently; 4 = always or almost always. Higher scores indicated a higher level of loneliness.

Sociodemographic variables such as age, gender, diagnosed medical conditions (i.e., diabetes, hypertension, cancer, among others) and whether the participant lived with children (under 18 years old) were self-reported and used in the study (see Table 1).

### 2.5. Data Analysis

Data analysis was performed using Mplus 8.1 [34] using full information maximum likelihood (FIML) estimation. This method uses all available data to estimate model parameters, including information from participants who only provide data on the first occasion [35,36,37]. Latent growth curve modeling (LGCM) analysis was carried out to assess how unhealthy food and alcohol consumption behaviors change over time using a within-subject approach. LGCM is a structural equation modeling technique for describing change over time using repeated measures of variables as a function of time and, if incorporated, predictors. These repeated measures were modeled in latent variables representing the intercept of the construct (i.e., the starting point) and the slope, which describes the rate of change over time (i.e., trajectory) [38].

Five models were tested to assess our hypothesis: (a) an unconditional linear model, specifying the intercept and the linear slope of unhealthy food and alcohol consumption, (b) an unconditional quadratic conditional, defining the intercept and a linear and quadratic slope and, (c) a conditional model including the following predictors measured at W1: age, perceived COVID-19 economic impact, COVID-19 interpersonal impact, COVID-19 health impact, depression, stress, anxiety and loneliness. Significant predictors of the intercept term explained why people varied in unhealthy food and alcohol consumption at W1. In contrast, predictors of the linear slope explained why individuals differed in their rates of change. Finally, a conditional model was tested using gender (Model d) and living with children or without children (Model e) as moderators.

All models were assessed using the comparative fit index (CFI), the Tucker-Lewis index (TLI), the standardized root mean square residual (SRMR), the root mean square error of approximation (RMSEA) with a confidence interval of 90%, the Akaike information criterion (AIC), the Bayesian information criterion (BIC) and the sample-size adjusted Bayesian information criterion (S-BIC). These indices were interpreted according to the goodness-of-fit criteria: CFI and TLI > 0.95, SRMR and RMSEA ≤ 0.08 and lower values of AIC, BIC and S-BIC, indicated a good model fit [39,40].

## 3. Results

### 3.1. General Results

In the first stage of the analysis, an unconditional linear model was estimated by specifying the repeated measures of unhealthy food and alcohol consumption as an intercept and linear slope function. Results for this model (Model a) showed an adequate model fit, *χ*^2^(43) = 129.562, *p* < 0.001, CFI = 0.945, TLI = 0.942, RMSEA = 0.045 [0.036, 0.054], SRMR = 0.048. As shown in Table 2, standardized parameter estimates for Model 1 display evidence of significant variability on the intercept of unhealthy food consumption, whereas there was a non-significant linear effect in the slope. This non-significant effect indicates that, for this sample, there were no changes in the consumption of unhealthy food over time. A significant and negative linear trajectory effect was found for alcohol consumption, describing gradual decreases in alcohol consumption over time for the sample.

A second model included an additional quadratic term (Model b). As there was not enough variability in the quadratic term, this was fixed to 0 as suggested by the estimation process. Model b showed good overall model fit, *χ*^2^(41) = 105.475, *p* < 0.001, CFI = 0.959, TLI = 0.955, RMSEA = 0.039 [0.030, 0.049], SRMR = 0.048. Despite the good model fit indices, the overall quadratic effect was not significant (*p* > 0.05). Therefore, it was decided to retain the more parsimonious linear model, which was used for estimating the following conditional model (Model c).

Considering that model estimates indicated significant variance around the mean for both intercept and linear slope (only for alcohol consumption), baseline (baseline refers to the measures completed at W1.) predictors were introduced in a conditional model. This model (Model c) showed an acceptable model fit, *χ*^2^(103) = 237.884, *p* < 0.001, CFI = 0.942, TLI = 0.919, RMSEA = 0.038 [0.032, 0.044], SRMR = 0.036 (see Table 3). In this model, it was found that the slope of unhealthy food consumption and the slope of alcohol consumption were positively related, indicating that as the consumption of unhealthy food increases over time, so does the consumption of alcohol. Results suggested a negative covariance between the alcohol consumption intercept and alcohol consumption slope, indicating that participants with higher alcohol consumption at baseline showed rapid decreases in alcohol consumption over time.

#### 3.1.1. Predictors for Unhealthy Food Consumption

As shown in Table 3, age was a significant predictor of the intercept, indicating that older age had lower consumption of unhealthy food. In contrast, perceived impact of COVID-19 on health and interpersonal relations was a significant and marginally significant predictor of the baseline levels of unhealthy food consumption, indicating that participants who had a higher perceived impact of COVID-19 had higher unhealthy food consumption. In the linear slope, only depression was a significant predictor, indicating that participants with higher levels of depression at baseline predicted more rapid decreases in unhealthy food consumption over time.

#### 3.1.2. Predictors for Alcohol Consumption

As Table 3 suggested, age and perceived COVID-19 economic impact were related to baseline scores, indicating that older participants and participants who had suffered more economically had higher levels of alcohol consumption at baseline. In the linear slope case, age was also a significant predictor, indicating that older participants reported more rapid alcohol consumption increases over time.

### 3.2. Moderated LGC Modeling

#### 3.2.1. Gender Moderation

As the conditional model (Model c) presented a good model fit, a moderated conditional model was estimated to assess if gender interacts with the other predictors of the growth over time of unhealthy food and alcohol consumption. This moderated LGCM showed an acceptable model fit, *χ*^2^(194) = 392.656, *p* < 0.001, CFI = 0.920, TLI = 0.889, RMSEA = 0.048 [0.041, 0.054], SRMR = 0.053. Standardized parameter estimates are given in Table 4. This model showed that, at baseline, women consumed more unhealthy food and alcohol than men (intercept = 2.091 for women vs. 1.842 for men). The slopes were not significant for unhealthy food, meaning no changes occurred over time (*p* > 0.05). However, there was a negative linear effect for the slope in alcohol consumption, describing a gradual decrease in alcohol consumption over time in both genders (slope = −0.841 for women vs. −0.742 for men). Moreover, women showed higher initial alcohol consumption levels than men at baseline (intercept = 0.533 for women vs. 0.243 for men).
Predictors of unhealthy food consumption in women and men: Age was significantly and negatively related to unhealthy food consumption at baseline, indicating that older participants had lower initial levels of unhealthy food consumption. Meanwhile, perceived COVID-19 interpersonal impact was significantly associated with higher initial levels of unhealthy food consumption for women but not men. Table 3 also shows the predictors of the linear slope of unhealthy food consumption. For women only, depression was significantly related to rapid decreases in unhealthy food consumption over time.Predictors of alcohol consumption in women and men: For men only, perceived COVID-19 economic impact was significantly and positively related to higher alcohol consumption at baseline. Age was positively correlated with higher alcohol consumption in men. Concerning the slope coefficient, it was found that perceived COVID-19 economic impact at baseline predicted more rapid decreases in alcohol consumption in men, but not in women. For both women and men, age was a significant predictor, indicating that older participants reported more rapid alcohol consumption increases over time. For women only, loneliness at baseline predicted a rapid decrease in alcohol consumption over time.

#### 3.2.2. Living with or without Children Moderation

Finally, one of the aims of this study was to evaluate whether living with children was also a moderator in women and men. An LGCM was estimated using only the women participants and the variable living with or without children as a moderator. This model showed an acceptable fit, *χ*^2^(194) = 304.892, *p* < 0.001, CFI = 0.932, TLI = 0.905, RMSEA = 0.042 [0.033, 0.051], SRMR = 0.062. In contrast and probably because of the small sample size, this same model could not be estimated for men. Therefore, only the moderated model for women is presented in Table 4. This model shows that, at baseline, women not living with children consumed more unhealthy food and alcohol than women living with children (Food_i_ = 1.879 for living with children vs. 2.223 for living without children; Alcohol_i_ = 0.137 for women living with children vs. 0.806 for women living without children). Similarly, to previous models, the slopes were not significant for either group (*p* > 0.05).
Predictors for unhealthy food consumption in women who live and do not live with children: As in previous results, age was significantly and negatively related to women’s unhealthy food consumption. For women not living with children, anxiety was a predictor of higher levels of unhealthy food consumption at baseline. Importantly, it was found that for women living with children only, perceived interpersonal impact of COVID-19 and stress were positively associated with higher levels of unhealthy food consumption. Regarding the slope coefficient, women not living with children who had higher levels of depression showed more rapid decreases in unhealthy food consumption over time.Predictors for alcohol consumption in women who live and do not live with children: For women living with children, perceived COVID-19 economic impact and age were significantly related to higher alcohol consumption levels at baseline. This parameter indicates that more economically affected and older women living with children reported higher alcohol consumption levels at the start. Older women participants (living with and without children) reported more rapid increases in alcohol consumption time. For women living with children, anxiety and loneliness were associated with the slope coefficient. Participants with higher anxiety levels at baseline showed a rapid increase in alcohol consumption over time. In the case of loneliness, the association was negative, indicating that higher levels of loneliness at baseline were associated with a rapid decrease in alcohol consumption over time.

## 4. Discussion

The results obtained by the present study are an essential foundation for advancing our understanding of how food and alcohol consumption have changed longitudinally during the pandemic in university students and staff, identifying which variables play a role in explaining such change and, finally, whether these predictors act in the same way for women vs. men and women living with or without children.

The results revealed that although higher levels of unhealthy food consumption were presented at the start of the pandemic, there were no changes in the consumption of unhealthy food over time; instead, it remained steadily high. These results are consistent with other studies that have found an unhealthy food consumption pattern during COVID-19 confinement in Europe, North Africa, Western Asia and the Americas [41]. Moreover, eating unhealthy food could be attributed to anxiety, boredom, or emotional eating [42,43]. Concerning alcohol consumption, we identified a gradually decreasing trajectory, implying that participants, on average, reduced their alcohol consumption as time passed. The decrease in alcohol consumption might be explained by restrictions on buying non-essential goods, such as alcohol due to the economic crisis and instability generated by the pandemic [6,44]. Moreover, the recommendations of social distancing have reduced the possibilities for social interactions, which is one of the main motives for drinking [45,46,47].

We also found that younger participants and those who perceived a higher COVID-19 health impact predicted higher initial levels of unhealthy food consumption during the pandemic. At the same time, higher levels of depression predicted a decrease in food consumption over time. Although previous research has suggested that depression is associated with eating behaviors (mainly overeating) [9], it is likely that work overload and the consequent lack of time produced by the pandemic is acting as a moderator and buffers the impulse to eat [19]. Thus, participants could have less time available and more disposition to cook unhealthy food. It is also possible that this result is biased, because in our survey we only asked for information on unhealthy food intake, not all the food consumed. This could mean that overall food intake and not just unhealthy food consumption was reduced during the time we collected our data. Future research should consider this limitation.

Regarding alcohol consumption, older participants were associated with more alcohol consumption at the baseline and over time. It is likely, as Chodkiewicz and colleagues (2020) [48] suggest, that older participants engaged in more alcohol consumption as a reward due to the lack of other sources of reward (i.e., social rewards) [30]. In addition, these results could be explained as a coping mechanism to deal with long working hours [9] or as a way to separate working and non-working hours. This result was found for participants regardless of gender and living with or without children.

Furthermore, we identified a moderator effect of gender and living with or without children in women. This moderation implies that different variables predicted trajectories of unhealthy food and alcohol consumption. Specifically, perceived COVID-19 interpersonal impact predicted more unhealthy food consumption over time for women (vs. men) and for women living with children. In the case of women not living with children, higher scores of anxiety and higher perceived COVID-19 interpersonal impact predicted more unhealthy food consumption at the beginning. This pattern of unhealthy food consumption is possibly a coping mechanism to deal with higher demands related to having less social, institutional, or educational support available (i.e., nurseries, schools, domestic help, among others) [18,49]. These difficulties are associated with higher stress and anxiety, especially during the first months of the pandemic, leading to more emotional eating [43].

Consequently, for all participants, youth was associated with more unhealthy food consumption over time. Previous research by Schnettler et al. (2015), Li et al. (2012) and Verger et al. (2009) [50,51,52] has found that when students become responsible for their own meals, they engage in riskier eating behaviors such as eating food with high energy density or fast food, because of hurried lifestyles, lack of time for food selection, organization, preparation and intake. Regarding the trajectory of unhealthy food consumption, for women only (vs. men) and women not living with children, higher depression scores were associated with lower unhealthy food consumption over time. Depression is usually associated with loss of appetite [53]. This means that persons with higher depression scores are reported to show less concern about food preparation and a markedly diminished interest or pleasure in all or almost all, everyday activities [54,55], which could explain our findings.

We also showed that higher anxiety levels and higher perceived COVID-19 economic impact predicted more alcohol consumption at the baseline for men. In this study, men showed a different pattern in alcohol consumption than the total sample. This may be because in Chile, traditional gender roles dictate that men are still the financial providers, which, in an economic crisis, is associated with anxiety, fear and concern arising from uncertainty [6]. Concerning alcohol consumption across time, we found that higher loneliness scores predicted less alcohol consumption longitudinally for women and women living with children, which seems counterintuitive [56]. Women who feel alone, possibly, do not engage in alcohol consumption because they lack the opportunities to use it as a social facilitator (i.e., during lockdown) [57] or, they participate in different coping strategies not associated with drinking or eating. These aspects must be considered in future studies. Finally, independently of living with or without children, older women were associated with more alcohol consumption over time. However, our sample was mainly comprised of university students, for whom alcohol might be less available and affordable than for older women [30,58].

The strengths of this study include detailed longitudinal data (monthly) on trajectories of eating and drinking behavior during the pandemic. In addition, the sample is heterogeneous, including participants in different stages of their life cycle and therefore has potentially generalizable results. Our findings allow us to identify specific predictors, especially for high-risk groups such as women and women living with children. Further, our findings on predictors of alcohol and food consumption, show that interventions should not only be aimed at reducing the effect of the pandemic on food and alcohol consumption, but should also consider variables associated with emotional distress.

### Limitations

Despite the novelty of this study, it does present several limitations. First, the study was based on an online survey. Although this is the best method of collecting data during the pandemic, our results only represent the behaviors reported by student and university staff. Moreover, participants without home access to the internet (e.g., service staff of the universities), who are probably the most affected by the pandemic, were not fully represented. Second, our study’s timeframe evaluates the participants’ risk behaviors at the time when Chile reported the highest numbers of COVID-19 infections during 2020 (W1). Although our study only followed up participants for a few months, these months are significant for the Chilean population because the beginning of the study was just a month after the peak of the pandemic in Chile and, specifically for university students and staff, this period corresponded to halfway through the academic year. Therefore, the results found here are limited to the particular context of those months. Third, our sample consisted mainly of women, which made it difficult to conduct a multigroup analysis comparing men living with or without children. Fourth, we did not have pre-pandemic data that permitted us to compare previous and new health risk behaviors and, therefore, establish a proper baseline. As a result, our analysis relies on retrospective data based on recall bias and this should be taken into consideration. Most of our limitations could be overcome using a random population sample and telephone surveys, especially for those with less access to, or familiarity with, online surveys. Future studies should consider assessing other health-risk behaviors such as marijuana, tobacco and opioid use.

## 5. Conclusions

Overall, these findings suggest that the pandemic has a high impact on health-risk behaviors beyond the contagion threat. That said, the effects of the pandemic should be addressed to prevent the consequences of unhealthy food and alcohol consumption over time. Any prevention strategy should take into account gender differences and the burden placed on women who live with children.

Finally, it is also essential to accept that the pandemic and its effects will not be over soon. Hence, it is necessary to continue assessing its longitudinal consequences on health risk behaviors, such as consuming unhealthy food or alcohol, over time.

## Figures and Tables

**Table 1 ijerph-18-04542-t001:** Sociodemographic characteristics of participants at W1 (*n* = 1038).

Characteristics	Total Sample *n* = 1038	Students *n* = 689	University Staff *n* = 349
Gender (women)	69%	70.8%	65.4%
Mean age (SD)	29.52 (11.66)	23.10 (4.53)	42.18 (11.06)
Educational level			
Higher education (university, tertiary)	53.5%	61.2%	38.5%
Post-graduate (masters, doctoral degree).	20.9%	2.1%	58%
Living alone	6.1%	4.1%	10%
Living with children	40.85%	37.59%	47.28%
Mean tally of diagnosed medical conditions (diabetes, depression, anxiety, cancer, fibromyalgia, others, cardiac and respiratory problems)	1.14	1.13	1.18

**Table 2 ijerph-18-04542-t002:** Parameter estimates and model fit for the unconditional and conditional latent growth curve modeling analyses of unhealthy food and alcohol consumption.

	Model a	Model b	Model c
	Unconditional Linear Model	Unconditional Quadratic Model	Conditional Linear Model
Food			
Mean			
Intercept _(i)_	9.124 **	8.972 **	
Linear Slope _(ls)_	0.077	0.424 **	
Quadratic slope _(qs)_		−0.087	
Variance			
Intercept _(i)_	21.159 **	21.242 **	
Linear Slope _(ls)_	0.329 **	0.338 **	
Quadratic slope _(qs)_		0	
Alcohol			
Mean			
Intercept _(i)_	2.351 **		
Linear Slope _(ls)_	−0.147 **		
Quadratic slope _(qs)_	-		
Variance			
Intercept _(i)_	7.628 **	7.654 **	
Linear Slope _(ls)_	0.664 **	0.662 *	
Quadratic slope _(qs)_		0	
Covariances			
Food _(i)_ − Alcohol _(i)_	−0.070	−0.061	−0.051
Food _(ls)_ − Alcohol _(ls)_	0.307 *	0.255	0.323 *
Food _(i)_ − Food _(ls)_	−0.096	−0.096	−0.097
Alcohol _(i)_ − Alcohol _(ls)_	−0.247 **	−0.639 *	−0.257 *
Model fit			
AIC	37,166.526	37,132.927	34,868.444
BIC	37,274.694	37,250.929	35,167.149
S-BIC	37,204.820	37,174.703	34,970.246

Note. ** *p* < 0.01 * *p* < 0.05, _i_ = intercept, _ls_ = linear slope, _qs_ = quadratic slope, AIC, Akaike Information Criterion; BIC, Bayesian Information Criterion; S-BIC, Sample-size adjusted Bayesian Information Criterion.

**Table 3 ijerph-18-04542-t003:** Parameter estimates for the conditional linear model (Model c) for all participants.

	Food		Alcohol	
	Intercept R^2^ = 0.169	Slope R^2^ = 0.110	Intercept R^2^ = 0.021	Slope R^2^ = 0.045
Age	−0.214 **	0.004	0.128 **	0.179 **
COVID-19 economic impact	0.047	0.006	0.068	−0.055
COVID-19 health impact	0.108 **	−0.130	0.028	−0.009
COVID-19 interpersonal impact	0.081	0.066	−0.015	0.080
Depression	−0.008	−0.300 *	0.072	0.029
Anxiety	0.062	0.103	0.020	0.014
Stress	0.093	0.117	−0.017	0.068
Loneliness	0.027	−0.159	−0.025	−0.047
Living with children	0.028	0.024	−0.010	−0.032

Note. ** *p* < 0.01 * *p* < 0.05. Information regarding model fit and covariances is shown in Table 2. R^2^ = represents the proportion of the variance in the outcomes that the predictors explain conjointly.

**Table 4 ijerph-18-04542-t004:** Parameter estimates for moderated latent growth curve modeling, using women and men and living with or without children.

	Women (*n* = 637) vs. *Men* (*n = 269*)				Women Living with Children (*n* = 370) vs. *Living without Children* (*n* = 267)			
	Alcohol Intercept	Alcohol Slope	Food Intercept	Food Slope	Alcohol Intercept	Alcohol Slope	Food Intercept	Food Slope
Age	0.119 ^†^	**0.204** *	**−0.237** **	0.055	**0.298** **	**0.233** ***	−0.180 *	0.053
	***0.164*** **	***0.202 ****	***−0.194* ****	*0.016*	*0.039*	***0.202*** *******	*−0.287* **	*0.102*
COVID-19 economic impact	0.035	−0.010	0.061	−0.086	0.136 ^†^	−0.011	0.100	−0.085
	***0.114*** *	*−0.118* ^†^	*0.006*	*0.158*	*−0.009*	*−0.030*	*0.021*	*−0.131*
COVID-19 health impact	−0.057	0.120	0.072	−0.037	−0.093	0.121	0.095	−0.085
	*0.046*	*0.011*	*0.097*	*0.215*	*−0.033*	*0.145*	*0.054*	*−0.032*
COVID-19 interpersonal impact	0.030	−0.055	**0.142** **	−0.109	0.023	−0.039	0.210 *	−0.260
	*0.034*	*0.048*	*0.071*	*−0.042*	*0.021*	*−0.029*	*0.075*	*−0.252*
Depression	0.044	−0.001	−0.050	**−0.558** *	0.000	0.114	0.075	−0.355
	*0.062*	*0.195*	*0.053*	*0.054*	*0.084*	*−0.180*	*0.002*	***−0.661*** *
Anxiety	−0.085	0.108	0.123	0.020	−0.031	0.252 ^†^	−0.085	0.678
	***0.244*** **	*−0.208*	*−0.057*	*−0.207*	*−0.108*	*−0.109*	*0.279 ***	*−0.166*
Stress	0.122	0.157	0.054	0.293	0.194	0.072	0.241 ^†^	0.534
	*0.185* ^†^	*−0.058*	*0.156*	*0.184*	*0.010*	*0.285*	*−0.073*	***0.854*** *
Loneliness	0.015	**−0.168** *	0.035	−0.185	0.059	**−0.217** *	−0.008	−0.026
	*−0.054*	*0.108*	*0.018*	*−0.081*	*0.020*	*−0.014*	*0.064*	*−0.227*

Note. Italics indicates the parameter estimates for Men and for Women Living without Children. Significant results are in bold, ** *p* < 0.01, * *p* < 0.05, ^†^
*p* < 0.10.

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
