# Peer review of "COVID-19 Perceived Impact and Psychological Variables as Predictors of Unhealthy Food and Alcohol Consumption Trajectories: The Role of Gender and Living with Children as Moderators"

_ijerph, 2021, doi:10.3390/ijerph18094542_

Round 1

Reviewer 1 Report

This manuscript describes an important and timely topic, specifically the role of psychological distress and impact of COVID on economic, health, and interpersonal domains in trajectories of unhealthy food and alcohol consumption. The authors present detailed analyses utilizing structural equation models that help to characterize effects of pandemic stress on health behaviors. The authors also examine moderators of gender and living with children. Here are some suggestions for revising the manuscript:

-Abstract sentence on “aimed at minimizing the impact of the pandemic on unhealthy food consumption over time” should add “and alcohol” given dual focus.

-The opening of the introduction discusses harmful alcohol use as the leading cause of death and disability worldwide. My understanding is that this is not accurate – that cardiovascular disease is the leading cause of death and disability worldwide. WHO reference listed does not seem to support the author’s statement. Please verify and edit for accuracy.

-Response rates listed in Methods are unclear – I am assuming the authors mean that 50.67% completed at wave 1 of all those invited to participate but then those percentages listed for waves 2-4 should be clarified – are these for all those invited or all those that completed wave 1?

-Please clarify whether description of a standard alcoholic drink was provided to participants so that they were using same metric for responding to the single item measure of number of drinks consumed in the past week. If this was not provided, this should be added to limitations.

-Similarly more information should be provided about unhealthy food consumption measurement. Is there evidence that people are reliable in reporting on food consumption occurring as long as 7 days ago? Is snacking a common metric of unhealthy food consumption (since description given does not specify types of snacks and most dietitians recommend eating 2 snacks daily between meals)?

-Some places in the manuscript refer to overeating and others unhealthy eating so recommend editing this to be consistent with what was measured.

-Please include Cronbach’s alpha for scales used in the study to assess reliability.

-Were depression, stress, anxiety, COVID impact measured at each time point or only at baseline? If these variables were measured longitudinally, this could really add to the analysis to examine change in these variables in association with change in outcomes of interest.

-Discussion: Additional consideration should be given to this specific timeframe assessed and how that might impact results/what might be expected if wider timeframe was used. Limitation that the data is across a few months mid-pandemic.

-The sentences beginning with “Regarding the trajectory of unhealthy food consumption… lower depression scores were associated with higher unhealthy food consumption over time… which could explain our findings” does not make sense. The authors state that less depressed people had higher food consumption, and then justify by saying more depressed people increase carb consumption and this could explain findings.

-There are typos (e.g., “adaptative”) throughout the manuscript that should be carefully edited. -There are a couple of places in which the same result is repeated twice (e.g., “This model showed that at baseline women consumed more unhealthy food and alcohol than men” with stats cited for unhealthy food and then finding repeated at end of same paragraph referring to alcohol; same issue a few paragraphs down in describing effects for women consuming more alcohol and food if not living with children).

Author Response

-Abstract sentence on “aimed at minimizing the impact of the pandemic on unhealthy food consumption over time” should add “and alcohol” given dual focus.

            Response: Thank you, we added your suggestion to the abstract (see line 25).

-The opening of the introduction discusses harmful alcohol use as the leading cause of death and disability worldwide. My understanding is that this is not accurate – that cardiovascular disease is the leading cause of death and disability worldwide. WHO reference listed does not seem to support the author’s statement. Please verify and edit for accuracy.

            Response: Thank you. According to the OMS, “Alcohol consumption contributes to 3 million deaths each year globally as well as to the disabilities and poor health of millions of people. Moreover, harmful use of alcohol is responsible for 5.1% of the global burden of disease”.

We changed the sentence to (see lines 31-33):

The prevalence of unhealthy food consumption and high alcohol consumption is growing at an alarming rate in the Americas [1]. Alcohol is the most used toxic substances worldwide and its harmful consumption causes several disabilities and even death, while obesity and overweight are the second leading cause of preventable deaths.

-Response rates listed in Methods are unclear – I am assuming the authors mean that 50.67% completed at wave 1 of all those invited to participate but then those percentages listed for waves 2-4 should be clarified – are these for all those invited or all those that completed wave 1?

            Response: Thank you for this comment. We deleted the percentages and included the exact sample size for the participants at each wave. In the text (see lines 99-100):

The current study included 1038 participants who reported their alcohol and unhealthy food consumption behaviors over four waves during July (Wave 1, n = 1038), August (Wave 2, n = 509), September (Wave 3, n = 413) and October (wave 4, n = 475).

-Please clarify whether description of a standard alcoholic drink was provided to participants so that they were using same metric for responding to the single item measure of number of drinks consumed in the past week. If this was not provided, this should be added to limitations.

            Response: Thanks. We added this information on the methods section. In the text (see lines 152-159)

Past-week alcohol consumption

Participants were asked, “In the past week, how many drinks did you consume?”. Responses were scored on a scale from 0 to 20 drinks. Participants were provided with information about standard alcoholic drinks, using images and giving the volume of an equivalent amount of beer, wine, and spirits. They had to report the number of drinks consumed yesterday and during the previous week (7 days).

 -Similarly, more information should be provided about unhealthy food consumption measurement. Is there evidence that people are reliable in reporting on food consumption occurring as long as 7 days ago? Is snacking a common metric of unhealthy food consumption (since description given does not specify types of snacks and most dietitians recommend eating 2 snacks daily between meals)?

            Response: In the Method section we added the examples given to the participants referring to these types of unhealthy food consumption (see lines 160-169). It now reads:

Past-week unhealthy food consumption

A 5-item question asked, "During the last week, on how many days have you consumed: fried meals (e.g., fried meat, fish, eggs, fries), sugary drinks (i.e., cola drinks, bottle juice), desserts or candies (e.g., ice cream, chocolate, candies, cakes, pastries), snacks (potato chips, chocolate bars, candy bar, cookies), and fast food (e.g., hamburgers, pizzas, hot dogs)." All these questions refer to foods that are highly processed and rich in saturated fat, sugar and sodium. Responses were scored from 0 to 7 days. Responses were summed to provide a total score indicating comfort food consumption. High scores indicated greater unhealthy food consumption. A reliability analysis showed acceptable levels ( = 0.768).

The snacks that we provided for the participants as examples of unhealthy snacking would not qualify as the snacks recommended by dietitians. We do not have evidence that people can reliably report on food consumption as long as 7 days ago, therefore it was included as a limitation (see lines 594-595 on Limitations).

-Some places in the manuscript refer to overeating and others unhealthy eating so recommend editing this to be consistent with what was measured.

            Response: As suggested, we have corrected this throughout the manuscript (see lines 79, 432, 498, and 527-528).

-Please include Cronbach’s alpha for scales used in the study to assess reliability.

            Response: We have included Cronbach’s alpha for the scales used in the study (see lines 159, 168-169, 228-229).

-Were depression, stress, anxiety, COVID impact measured at each time point or only at baseline? If these variables were measured longitudinally, this could really add to the analysis to examine change in these variables in association with change in outcomes of interest.

            Response: We appreciate your comment and we thought the analysis suggested could be very interesting. Although we assessed these variables longitudinally, our study aims to evaluate the trajectories over time of the alcohol and unhealthy food consumption.

-Discussion: Additional consideration should be given to this specific timeframe assessed and how that might impact results/what might be expected if wider timeframe was used. Limitation that the data is across a few months mid-pandemic.

            Response: Thank you for pointing this out. We have added this to the limitations of our study. The revised text reads as follow (see lines 584-590):

    Second, our study's timeframe evaluates the participants' risk behaviors, at the time when Chile reported the highest numbers of COVID-19 infections during 2020 (W1). Although our study only followed up participants for a few months, these months are significant for the Chilean population because the beginning of the study was just a month after the peak of the pandemic in Chile and, specifically for university students and staff, this period corresponded to halfway through the academic year. Therefore, the results found here are limited to the particular context of those months.

-The sentences beginning with “Regarding the trajectory of unhealthy food consumption… lower depression scores were associated with higher unhealthy food consumption over time… which could explain our findings” does not make sense. The authors state that less depressed people had higher food consumption, and then justify by saying more depressed people increase carb consumption and this could explain findings.

            Response: Thank you for pointing this out. We realize that the way we reported the association was odd and we made it more straightforward. We also clarified our findings about this statement. The revised text read as follows (see lines 525-531):

Regarding the trajectory of unhealthy food consumption, for women only (vs. men) and women not living with children, higher depression scores were associated with lower unhealthy food consumption over time. Depression is usually associated with loss of appetite. This means that persons with higher depression scores are reported to show less concern about food preparation and a markedly diminished interest or pleasure in all or almost all, everyday activities, which could explain our findings.

-There are typos (e.g., “adaptative”) throughout the manuscript that should be carefully edited.

Response: Thank you. As suggested, typos were corrected (see line 60).

-There are a couple of places in which the same result is repeated twice (e.g., “This model showed that at baseline women consumed more unhealthy food and alcohol than men” with stats cited for unhealthy food and then finding repeated at end of same paragraph referring to alcohol; same issue a few paragraphs down in describing effects for women consuming more alcohol and food if not living with children).

            Response: Thank you for your comments. We addressed the issues with repeated results. The revised text reads as follows: (see lines 353-359 for the first paragraph and lines 395-400 for the second paragraph):

This model showed that, at baseline, women consumed more unhealthy food and alcohol than men (intercept = 2.091 for women vs. 1.842 for men). The slopes were not significant for unhealthy food, meaning no changes occurred over time (p > 0.05). However, there was a negative linear effect for the intercept and the slope in alcohol consumption, describing a gradual decrease in alcohol consumption over time in both genders (slope = -0.841 for women vs. -0.742 for men). Moreover, women showed higher initial alcohol consumption levels than men at baseline (intercept = 0.533 for women vs. 0.243 for men). 

Therefore, only the moderated model for women is presented in Table 4. This model shows that, at baseline, women not living with children consumed more unhealthy food and alcohol than women living with children (Foodiintercept = 1.879 for living with children vs. 2.223 for living without children; Alcoholi = 0.137 for women living with children vs. 0.806 for women living without children). Similarly to previous models, the slopes were not significant for either group of women (p > 0.05). Nevertheless, there was a negative linear effect for both groups in alcohol consumption, describing a gradual decrease in alcohol consumption over time for the sample (slope = -0.964 for living with children vs. -0.953 for living without children). Women not living with children showed higher initial alcohol consumption levels than women living with children at baseline (intercept = -0.137 for women living with children vs. 0.806 for women living without children).

Reviewer 2 Report

Review of the manuscript: IJERPH-1160451

In this paper the authors report trajectories of alcohol consumption and unhealthy food consumption in a sample of university staff and students in Chile during the COVID-19 pandemic. The work adds important longitudinal evidence on the lifestyle impacts of the pandemic and pandemic related changes to the home environment and mental health.

I enjoyed reviewing this manuscript - it is of very good quality, well-structured and excellently written. The data was of high quality, included sound analyses and the authors did a brilliant job of explaining complex results. As both alcohol consumption as well as food consumption show regional differences, I believe the manuscript could be improved by referring to the study population and therefore generalisability of the findings more clearly throughout.

Main revisions / clarification required:

  1. Abstract: The abstract could be improved by adding information on where the data was collected. Also it would be helpful to clarify how “unhealthy food consumption” was defined.
  2. Page 2 response rates. Please clarify the denominator for response rates. Is 40.75% in W2 referring to all that participated in W1 or all invited previously, meaning the students and staff of the two Chilean universities.
  3. Table 1 refers to medical conditions, could the authors explain how these were collected and whether they refer to current or ever diagnosed conditions in the table?
  4. Could the authors clarify how living with children was defined? Was it only referring to children <18 years, or younger.
  5. In the beginning of the discussion the authors refer to the findings quite general. However, it has to be noted that the sample might not be representative for the whole population, as well as potentially specific to Chile. Considering that alcohol consumption pattern vary by country/region as well as socio-economic status - the authors could improve the manuscript by clarifying that the results refer to university staff and students in Chile.
  6. When discussing the association of depression and eating behaviours line 303-309 the authors mainly argue with overeating, but in line 335 they reference association with both over and undereating. Furthermore appetite loss has been described as a symptom of depression. Since the authors do not have overall food consumption data it is possible that participants reduced their overall food intake including unhealthy foods. The authors might consider discussing this in lines 303-309.
  7. In line 350 the authors suggest women might cope in other ways than drinking or eating, but in line 356 they suggest more frequent and heavier drinking as a likely coping strategy in women. Could the authors clarify this inconsistency?
  8. In the limitations there is missing reference to the generalisability of the results and representativeness of the sample. Could the authors add this context?

Minor revisions:

  • Abstract: Clarify that moderation of living with and without children is referring to women only.
  • Page 2 add names to the Chilean universities
  • Line 220 and line 224/225 – is the information about alcohol intake repeated here? If not, how do they differ?
  • Line 310 – Grammatical error “older participants were” should be “older age was”

Author Response

Response: Many thanks for your comments.

Main revisions / clarification required:

  1. Abstract: The abstract could be improved by adding information on where the data was collected. Also it would be helpful to clarify how “unhealthy food consumption” was defined.

Response: We added the information to the abstract. Now the abstract reads as follows (see lines 15-16):

Data were collected through online surveys administered to 1038 participants from two universities (staff and students) in Chile, across five waves (July to October 2020). Participants provided information about their past-week unhealthy food and alcohol consumption, and mental health.

Regarding unhealthy food consumption, we clarified the term by using the examples given to the participants in the Method section (see lines 160-169).

2.3.2. Past-week unhealthy food consumption

A 5-item question asked, "During the last week, how many days have you consumed: fried meals (e.g., fried meat, fish, eggs, fries), sugary drinks (i.e., cola drinks, bottle juice), desserts or candies (e.g., ice cream, chocolate, candies, cakes, pastries), snacks (potato chips, chocolate bars, candy bar, cookies), and fast food (e.g., hamburgers, pizzas, hot dogs)." All these questions refer to foods that are highly processed and rich in saturated fat, sugar and sodium. Responses were scored from 0 to 7 days. Responses were summed to provide a total score indicating comfort food consumption. High scores indicated greater unhealthy food consumption. A reliability analysis showed acceptable levels (α = 0.768).

  1. Page 2 response rates. Please clarify the denominator for response rates. Is 40.75% in W2 referring to all that participated in W1 or all invited previously, meaning the students and staff of the two Chilean universities.

Response: Thank you for this comment. We deleted the percentages and included the exact sample size for the participants at each wave. In the text (see lines 99-100):

The current study included 1038 participants who reported their alcohol and unhealthy food consumption behaviors over four waves during July (Wave 1, n = 1038), August (Wave 2, n = 509), September (Wave 3, n = 413) and October (wave 4, n = 475).

  1. Table 1 refers to medical conditions, could the authors explain how these were collected and whether they refer to current or ever diagnosed conditions in the table?

Response: As suggested, we have added this information in lines 236-238, stating that these refer to diagnosed medical conditions.

In the manuscript:

Sociodemographic variables such as age, gender, diagnosed medical conditions (i.e., diabetes, hypertension, cancer, among others) and whether the participant lived with children (under 18 years old) were self-reported and used in the study (see Table 1).

  1. Could the authors clarify how living with children was defined? Was it only referring to children <18 years, or younger.

Response: We added this information in line 238, stating that children are under 18 years old. See previous response.

  1. In the beginning of the discussion the authors refer to the findings quite general. However, it has to be noted that the sample might not be representative for the whole population, as well as potentially specific to Chile. Considering that alcohol consumption pattern vary by country/region as well as socio-economic status - the authors could improve the manuscript by clarifying that the results refer to university staff and students in Chile.

Response: Thank you for your comment. We added your suggestion on the first lines of the discussion (see lines 422-426):

The results obtained by the present study are an essential foundation for advancing our understanding of how food and alcohol consumption have changed longitudinally during the pandemic in university students and staff, identifying which variables play a role in explaining such change, and finally, whether these predictors act in the same way for women vs. men and women living with or without children.

  1. When discussing the association of depression and eating behaviours line 303-309 the authors mainly argue with overeating, but in line 335 they reference association with both over and undereating. Furthermore appetite loss has been described as a symptom of depression. Since the authors do not have overall food consumption data it is possible that participants reduced their overall food intake including unhealthy foods. The authors might consider discussing this in lines 303-309.

Response: Thank you for pointing this out. We addressed your suggestion and made some changes to the manuscript. The revised text reads as follows (see lines 495-500):

Thus, participants could have less time available and more disposition to cook unhealthy food. It is also possible that this result is biased because in our survey we only asked for information on unhealthy food intake, not all the food consumed. This could mean that overall food intake and not just unhealthy food consumption was reduced during the time we collected our data. Future research should consider this limitation.

  1. In line 350 the authors suggest women might cope in other ways than drinking or eating, but in line 356 they suggest more frequent and heavier drinking as a likely coping strategy in women. Could the authors clarify this inconsistency?

Response: When referring to the ways in which women might cope other than drinking or eating, we are referring to women who feel alone. From line 539 onwards, we referred to older women. We clarify this point in the manuscript (see lines 539-546). The revised text reads as follows:

Women who feel alone, possibly, do not engage in alcohol consumption because they lack the opportunities to use it as a social facilitator (i.e. during lockdown) or, they participate in different coping strategies not associated with drinking or eating. These aspects must be considered in future studies. Finally, independently of living with or without children, older women were associated with more alcohol consumption over time. However, our sample was mainly comprised of university students, for whom alcohol might be less available and affordable than for older women.

  1. In the limitations there is missing reference to the generalisability of the results and representativeness of the sample. Could the authors add this context?

Response: Thank you for this comment. As suggested, we added a reference on the generalizability of our results. The text reads as follows (see lines 579-584):

Despite the novelty of this study, it does present several limitations. First, the study was based on an online survey. Although this is the best method of collecting data during the pandemic, our results only represent the behaviors reported by students and university staff. Moreover, participants without home access to the internet (e.g., the universities' service staff), who are probably the most affected by the pandemic, were not fully represented.

Minor revisions:

  • Abstract: Clarify that moderation of living with and without children is referring to women only.

Response: Thank you for the comment. We added this to the abstract. The revised text reads (see line 22):

Gender and living with or without children, for women only, were moderators of these trajectories.

  • Page 2 add names to the Chilean universities.

Response: Thank you for pointing this out, but the ethical committees that approved our project suggested avoiding naming the universities involved in our study to prevent participants from discrimination and stigmatization.

  • Line 220 and line 224/225 – is the information about alcohol intake repeated here? If not, how do they differ?

Response: In the revised manuscript, lines 353-354 refers to the intercept of unhealthy food consumption, and lines 358-359 refers to the intercept of alcohol consumption.

  • Line 310 – Grammatical error “older participants were” should be “older age was”

Response: Thank you, we corrected this error (see line 326).

Reviewer 3 Report

General comments:

Overall, I think this manuscript presents a well conducted study process. However, the foundation of the study (some of the survey items and assumptions) which cannot be changed, is problematic to me. 

  • The characterization of unhealthy eating: the question asks how many days have you consumed…[5 categories of food/beverages]. This strikes me as an inadequate approach to assessment; this data cannot adequately characterize unhealthy food.. E.g., depending on what foods are selected, daily snacks and daily desserts are not necessarily unhealthy. Fast food or fried foods might/might not be etc. The frequency (times/day) and volume (e.g., size of fried meals, sugary drink etc.) would have given much more information while still being fairly easy to collect.
  • The baseline: The first wave of data collection was July. By that time the pandemic/lock-down/social distancing protocols had been in place for several months. The pre-covid-19 pandemic period was, for most countries, prior to mid-March 2020. Yet, unless I am misinterpreting, the authors consider July 2020 the baseline. The loneliness item related to pre-pandemic – but this was the only one. But using July as baseline alters the interpretation of the data.
  • Question/item structure: For 3 of the items -- economic impact, health impact, and interpersonal impact, the questions (as presented) do not align with the response choice. Authors are asking a yes/no// true/false question and expect a participant to place this response on a 5-point scale.
    • Was the survey tested in the population prior to use? Was consideration given to participant interpretation of the questions – this was not mentioned in the methods.  

Specific comments:

Lines 69-70 – The statement that it is necessary to evaluate food/alcohol consumption patterns because the pandemic is long-lasting stressor is just not true.     

Line 85 – I’m assuming the authors mean routines or behaviors – but here, the term “habit” is not correct. 

Table 1

  • Title – are these just “baseline” characteristics? Did they change over the 4-month period?
  • Female sex: remove “of women” – unless you mean that 69% of women are in the Female sex category. For that matter, “sex” can be removed as well – it is redundant; “Female” is sufficient.
  • Authors posted religion characteristics – but did not have an option for “other” religion and did not use/refer to this information any place in the manuscript. Is it needed?
  • Are authors saying Mean # of total medical conditions? – either way, this should be made clear.

Line 139 – use of “habit” – instead of behaviors. As I read it, the authors did not measure habits.

Line 171 – here, and throughout the manuscript, the authors refer to a “non-gradual change” or to “non-gradual changes”. This term is confusing. As I reader I think am expecting large change if there was non-gradual; but it might also be used to refer to no change. Authors might figure out how to report their finding without this type of confusion/potential confusion.  

Table 2

Overall, this table is difficult (visually) to follow. Try re-spacing the headings (they run into each other) ; separating the sections; shading or otherwise separating columns; or whatever you can to make it easier for the reader to follow the content. 

Line 237 – it was either significant or it was not.  

Line 258-259 –  I think you mean: …”At baseline, women not living with children…”

Line 289-290 – a confusing use of the phrase “gradual change” (even though it is explained later in sentence).

Line 296-299 – “it is probable…” – this explanation is confusing.  I the introduction (lines 52-55) the Authors note that elevated stress could lead to increased alcohol consumption.

Line 324 – Authors relate situation to participant having “no social, institution, or educational support” But is that the case for these participants? i.e. do they have “no” …support?

Line 380/Conclusion paragraph

  • From what I can see, this study did not measure impact of pandemic on mental health.
  • The meaning of “addressed contingently” is not clear to me in this context. So, how will addressing the effects of the pandemic in and of itself, prevent unhealthy eating/obesity. Perhaps this part of the conclusion should be restated.
  • Also, eating food is not a health risk (it is a necessity). Perhaps you mean the consequences of consuming unhealthy foods or unhealthy amount of alcohol over time…as the authors, you can clarify based on what you mean.

Author Response

  • The characterization of unhealthy eating: the question asks how many days have you consumed…[5 categories of food/beverages]. This strikes me as an inadequate approach to assessment; this data cannot adequately characterize unhealthy food.. E.g., depending on what foods are selected, daily snacks and daily desserts are not necessarily unhealthy. Fast food or fried foods might/might not be etc. The frequency (times/day) and volume (e.g., size of fried meals, sugary drink etc.) would have given much more information while still being fairly easy to collect.

Response: We agree with you that volume would have given us much accurate information, but we believe that using the frequency of days on which people eat unhealthy food in the past week can be a proper proxy. We added more information in the methods section in which we characterize unhealthy food consumption using the examples provided to the participants of our study. All the examples given to the participants can be categorized as unhealthy food.  

In the manuscript (see lines 160-169):

2.3.2. Past-week unhealthy food consumption

A 5-item question asked, "During the last week, on how many days have you consumed: fried meals (e.g., fried meat, fish, eggs, fries), sugary drinks (i.e., cola drinks, bottle juice), desserts or candies (e.g., ice cream, chocolate, candies, cakes, pastries), snacks (potato chips, chocolate bars, candy bar, cookies), and fast food (e.g., hamburgers, pizzas, hot dogs)." All these questions refer to food that are highly processed and rich in saturated fat, sugar and sodium. Responses were scored from 0 to 7 days. Responses were summed to provide a total score indicating comfort food consumption. High scores indicated greater unhealthy food consumption. A reliability analysis showed acceptable levels ( α = 0.768).

  • The baseline: The first wave of data collection was July. By that time the pandemic/lock-down/social distancing protocols had been in place for several months. The pre-covid-19 pandemic period was, for most countries, prior to mid-March 2020. Yet, unless I am misinterpreting, the authors consider July 2020 the baseline. The loneliness item related to pre-pandemic – but this was the only one. But using July as baseline alters the interpretation of the data.

Response: We appreciate your comments. Perhaps we were not clear enough, but our study's baseline is not the baseline of the pandemic. To clarify this point, we add a footnote, noting that when we refer to baseline measures, we are referring to those measures assessed in W1 (see page 6).

  • Question/item structure: For 3 of the items -- economic impact, health impact, and interpersonal impact, the questions (as presented) do not align with the response choice. Authors are asking a yes/no// true/false question and expect a participant to place this response on a 5-point scale.

Response: Thank you for identifying this error. These three items were preceded by the following sentence: How much has the pandemic negatively affected you? The revised manuscript read as follows (see lines 171-223):

Also, we checked the translation of the response scale, and we modified it accordingly.  

2.4.1. Perceived COVID-19 impact

For the following three items the participants were asked to think about how much the COVID-19 pandemic has affected them.

2.4.1.1. Perceived COVID-19 economic impact. A single item question asked: “COVID-19 has negatively affected my economic or employment situation.” Responses were scored on a 5-point scale: not at all (0), a little (1), some (2), quite a bit (3), a lot (4). Higher scores indicated a greater economic impact of COVID-19.

2.4.1.2. Perceived COVID-19 health impact. A single item question asked: “COVID-19 has negatively affected me or my loved ones’ health”. Responses were scored on a 5-point scale (not at all (0), a little (1), some (2), quite a bit (3), a lot (4). Higher scores indicated a greater health impact of the COVID-19.

2.4.1.3. Perceived COVID-19 interpersonal impact. A single item question asked: “COVID-19 has negatively affected my personal relations with family and friends”. Responses were scored on a 5-point scale (not at all (0), A little (1), some (2), quite a bit (3), a lot (4)). Higher scores indicated a greater interpersonal impact of COVID-19.

    • Was the survey tested in the population prior to use? Was consideration given to participant interpretation of the questions – this was not mentioned in the methods.  

Response: Most of the instruments used in our survey have been validated for the Chilean population (i.e., DASS-21) and used nationally (i.e., Unhealthy food consumption items, alcohol items). Moreover, we conducted a pilot study to test the complete survey with participants of similar characteristics to our research, assessed by a panel of expert judges.

Specific comments:

Lines 69-70 – The statement that it is necessary to evaluate food/alcohol consumption patterns because the pandemic is long-lasting stressor is just not true.

                        Response: As suggested, we changed the wording of the statement and added a reference to support this claim (see lines 82-85):

Since the pandemic has been found to increase levels of stress across the population, it will be necessary to evaluate changes over time in health-risk behaviors and their predictors, that are known to be affected by stress.

Line 85 – I’m assuming the authors mean routines or behaviors – but here, the term “habit” is not correct. 

                        Response: Thank you for pointing this out. We made the suggested changes (see line 99)

The current study included 1038 participants who reported their alcohol and un-healthy food consumption behaviorsover four waves during July (Wave 1, n = 1038), August (Wave 2, n = 509), September (Wave 3, n = 413) and October (wave 4, n = 475). See Table 1 for a description of participant characteristics.

Table 1

  • Title – are these just “baseline” characteristics? Did they change over the 4-month period?

Response: Thank you for this comment. We changed the table title to sociodemographic characteristics. We only measured this data at wave 1. We assumed that if there were changes in the participant’s sociodemographics, they would be minimal, considering that our study's timeframe comprised only 4 months. The revised table is in line 102 (pages 2-3).

  • Female sex: remove “of women” – unless you mean that 69% of women are in the Female sex category. For that matter, “sex” can be removed as well – it is redundant; “Female” is sufficient.

Response: Thank you for pointing this out, we have changed the text accordingly. You can see this change in Table 1.

  • Authors posted religion characteristics – but did not have an option for “other” religion and did not use/refer to this information any place in the manuscript. Is it needed?
  •  

Response: As you point out, it is not needed and we have removed it.

  • Are authors saying Mean # of total medical conditions? – either way, this should be made clear.

Response: We added a line clarifying this information (see lines 236-238). In the revised text read as follows:

Sociodemographic variables such as age, gender, diagnosed medical conditions (i.e., diabetes, hypertension, cancer, among others) and whether the participant lived with children (under 18 years old) were self-reported and used in the study (see Table 1).

Line 139 – use of “habit” – instead of behaviors. As I read it, the authors did not measure habits.

            Response: Thank you, we changed “habit” to “behaviors” (see line 244). In the revised text read as follows:

     Latent growth curve modeling (LGCM) analysis was carried out to assess how unhealthy food and alcohol consumption behaviors change over time using a within-subject approach.

Line 171 – here, and throughout the manuscript, the authors refer to a “non-gradual change” or to “non-gradual changes”. This term is confusing. As I reader I think am expecting large change if there was non-gradual; but it might also be used to refer to no change. Authors might figure out how to report their finding without this type of confusion/potential confusion.  

            Response: Thank you for your suggestion. Throughout the manuscript we changed the term “non-gradual change/s” to “no change”, to clarify this point (see lines 288, 355, and 428).

Table 2

Overall, this table is difficult (visually) to follow. Try re-spacing the headings (they run into each other); separating the sections; shading or otherwise separating columns; or whatever you can to make it easier for the reader to follow the content. 

            Response: We agree with you and as suggested, we have changed this table and divided it into Table 2 and Table 3 (on pages 5 and 6, respectively, and in lines 292-293, and 320 in the revised manuscript).

Line 237 – it was either significant or it was not.  

            Response: As you suggested we deleted the sentence referring to marginally significant results (see lines 370-371).

              Age was positively correlated with higher alcohol consumption in men. (significantly) and women (marginally significantly). Stress was also a marginally significant predictor for men, indicating that higher stress levels at baseline were associated with higher alcohol consumption levels.

Line 258-259 –  I think you mean: …”At baseline, women not living with children…”

            Response: As also suggested by another reviewer, we have changed this paragraph as several lines were redundant. The revised text read as follows (lines 395-400):

Therefore, only the moderated model for women is presented in Table 4. This model shows that, at baseline, women not living with children consumed more unhealthy food and alcohol than women living with children (Foodiintercept = 1.879 for living with children vs. 2.223 for living without children; Alcoholi = 0.137 for women living with children vs. 0.806 for women living without children). Similarly to previous models, the slopes were not significant for either group of women (p > 0.05). Nevertheless, there was a negative linear effect for both groups in alcohol consumption, describing a gradual decrease in alcohol consumption over time for the sample (slope = -0.964 for living with children vs. -0.953 for living without children). Women not living with children showed higher initial alcohol consumption levels than women living with children at baseline (intercept = -0.137 for women living with children vs. 0.806 for women living without children).

Line 289-290 – a confusing use of the phrase “gradual change” (even though it is explained later in sentence).

            Response: Thank you, we have corrected this in response to a previous comments (see comment above referring to Line 171 and its response).

Line 296-299 – “it is probable…” – this explanation is confusing.  I the introduction (lines 52-55) the Authors note that elevated stress could lead to increased alcohol consumption.

            Response: Thank you, we have changed this sentence according to your suggestion. The revised text reads as follows (see lines 435-437):

The decrease in alcohol consumption might be explained by restrictions on buying non-essential goods, such as alcohol, due to the economic crisis and instability generated by the pandemic.

Line 324 – Authors relate situation to participant having “no social, institution, or educational support” But is that the case for these participants? i.e. do they have “no” …support?

            Response: Thanks for your suggestion. We have clarified this sentence. The revised text reads as follows (see lines 514-517):

This pattern of unhealthy food consumption is possibly a coping mechanism to deal with higher demands related to having less social, institutional, or educational support available (i.e., nurseries, schools, domestic help, among others).

Line 380/Conclusion paragraph

  • From what I can see, this study did not measure impact of pandemic on mental health.

Response: Thank you for pointing this out. We corrected the sentence. The revised text reads as follows (See lines 600-601):

Overall, these findings suggest that the pandemic has a high impact on health-risk behaviors beyond the contagion threat.

  • The meaning of “addressed contingently” is not clear to me in this context. So, how will addressing the effects of the pandemic in and of itself, prevent unhealthy eating/obesity. Perhaps this part of the conclusion should be restated.

Response: Thank you. We have clarified these sentences. The revised text reads as follows (See 601-604):

That said, the effects of the pandemic should be addressed to prevent the consequences of unhealthy food and alcohol consumption over time. Any prevention strategy should take into account gender differences and the burden placed on women who live with children.

  • Also, eating food is not a health risk (it is a necessity). Perhaps you mean the consequences of consuming unhealthy foods or unhealthy amount of alcohol over time…as the authors, you can clarify based on what you mean.

Response: As suggested, we have changed this sentence. The revised text reads as follows (See lines 605-607):

 Finally, it is also essential to accept that the pandemic and its effects will not be over soon. Hence, it is necessary to continue assessing its longitudinal consequences on health risk behaviors, such as consuming unhealthy food or alcohol, over time.